# Usefulness of atezolizumab plus bevacizumab as second-line therapy for patients with unresectable hepatocellular carcinoma

**Shinpei Yamaba, Yukinori Imai, Kayoko Sugawara, Yoshihito Uchida, Akira Fuchigami, Hiroshi Uchiya, Nobuaki Nakayama, Satoshi Mochida** *

Faculty of Medicine, Gastroenterology & Hepatology, Saitama Medical University, Iruma-gun, Saitama, Japan

* smochida@saitama-med.ac.jp

**Data Availability Statement:** All relevant data are within the manuscript and its Supporting Information files.

## Abstract

### Aim

To clarify the efficacy of atezolizumab (ATZ) plus bevacizumab (BEV) as the second-line therapy for patients with unresectable hepatocellular carcinoma (HCC).

### Methods

The subjects were 82 patients with HCC receiving ATZ/BEV, including 33 patients with previous therapies with molecular-targeted agents (MTA). Therapeutic efficacy was evaluated using contrast-enhanced CT according to the mRECIST.

### Results

The Child-Pugh scores were 5, 6,7 and 8 in 40, 35, 5 and 2 patients, respectively, and the extents of HCC progression were BCLC stage A, B and C in 3, 31 and 48 patients, respectively. Early therapeutic efficacy was evaluated in 67 patients, and percentages of patients achieving CR/PR/SD/PD until 12 weeks were 3.0%/29.9%/49.3%/17.9%, respectively, indicating ORR of 32.8% and DCR of 82.1%, The ORR was higher in MTA-naïve patients (40.5%) than in those after discontinuation of lenvatinib due to PD (7.7%, P = 0.0410), while the DCR was equivalent between both patients (83.3% vs 80.0%, P = 0.1184), and the multivariate analysis revealed previous MTA therapies with lenvatinib alone as a factor to deteriorate the ORR (HR of 4.846 (P = 0.0619)). The OS rates at 24 and 48 weeks were 86% and 72%, respectively, and the rates did not differ between MTA-naïve and MTA-experienced patients. Multivariate analyses revealed that achievement of CR, PR or SD and peripheral neutrophil/lymphocyte ratio were associated with a favorable outcome (HR of 0.124, P<0.0001 and 0.351, P = 0.0303).

### Conclusions

ATZ/BEV merits consideration even for MTA-experienced patients, since the OS was equivalent to those in MTA-naïve patients despite of an unfavorable early therapeutic efficacy.

**Funding:** Satoshi MOCHIDA has received speaking fees or honoraria from AbbVie GK, Gilead Sciences Inc., Otsuka Pharmaceutical Co., Ltd., Toray Medical Co. Ltd., Eisai Co., Ltd., ASKA Pharmaceutical Co., Ltd., has received research grants from EPS Corporation, Gilead Sciences Inc., MSD K.K., intellim Corporation, and scholarship grants from AbbVie GK, EA Pharma Co. Ltd., Eisai Co., Ltd., Sumitomo Pharma Co., Ltd. The funders had no role in study design, data collection and analysis, decision to publish, or preparation of the manuscript.

**Competing interests:** The authors have declared that no competing interests exist.

# Introduction

Hepatocellular carcinoma (HCC) is the fifth leading cause of cancer death in Japan [1]. While interventional radiology procedures, such as transcatheter arterial chemoembolization (TACE) and transcatheter arterial infusion chemotherapy (TAI) were commonly undertaken for patients with unresectable HCC, introduction of treatment with molecular-targeted agents (MTAs) markedly altered the therapeutic strategy for these patients [2]. In Child-Pugh class A unresectable HCC patients with extrahepatic metastasis, therapy with MTAs is recommended; lenvatinib as well as sorafenib are used as the first-line agents, and regorafenib as a second-line agent, according to the updated clinical practice guidelines published by the Japan society of Hepatology (JSH) in 2019 [2]. Subsequently, the IMbrave150 trial, a global phase III trial, revealed that both the overall survival (OS) rates and progression-free survival (PFS) rates in MTA-naïve patients with advanced HCC were significantly higher in those receiving combining immunotherapy plus chemotherapy with atezolizumab and bevacizumab (monoclonal antibodies binding to programmed cell death 1-ligand 1 (PD-L1) and vascular endothelial growth factor (VEGF)-A, respectively) than in those receiving chemotherapy with sorafenib alone [3]. Consequently, the Japanese clinical practice guideline updated in 2021 recommends atezolizumab plus bevacizumab as first-line therapy in Child-Pugh class A unresectable HCC patients suspicious for TACE refractoriness [4], similar to the guideline published by the American Association for the Study of Liver Diseases (AASLD) [5] and the American Society of Clinical Oncology (ASCO) [6].

The Japanese clinical practice guideline [4] also recommended sorafenib and lenvatinib as first-line MTAs for patients with unresectable HCC who are unsuitable candidates for atezolizumab plus bevacizumab therapy, such as patients with autoimmune diseases, and regorafenib, ramucirumab and cabozantinib as second-line MTAs, based on the results of phase III trials of these agents [7–9]. In clinical practice however, atezolizumab plus bevacizumab as well as lenvatinib has been used for unresectable HCC patients with a previous history of treatment with MTAs [10–12], even though the usefulness of these agents as second-line therapy is yet to be elucidated. Thus, in the present study, the significance of combining immunotherapy plus chemotherapy as second-line therapy was evaluated based on the outcomes of patients with unresectable HCC treated with atezolizumab plus bevacizumab in relation to the outcomes of previous treatment with MTAs.

# Patients and methods

## Patients and the study design

The subjects were 82 consecutive patients with unresectable HCC who received atezolizumab plus with bevacizumab therapy at the Saitama Medical University Hospital between December 2020 and August 2022. The demographic features, clinical characteristics and outcomes of the patients were evaluated retrospectively. The study was conducted with the approval of the Institutional Review Board of the Hospital (Byou 2022–085), and informed consent for the study from the patients was obtained via offering them the opt-out option.

The extents of liver damage in the subjects were assessed by determination of the Child-Pugh class, albumin-bilirubin (ALBI) scores, and the modified ALBI (mALBI) grades [13]; the extent of HCC progression was assessed according to the Barcelona Clinic Liver Cancer (BCLC) staging system [14].

## Evaluation of the efficacy and safety of combing immunotherapy with chemotherapy using atezolizumab plus bevacizumab

Patients were given atezolizumab at a dose of 1,200 mg and bevacizumab at dose of 15 mg/kg body weight intravenously every 3 weeks. Depending on the severity and nature of

the adverse events, either or both atezolizumab and bevacizumab was discontinued/ resumed.

The therapeutic efficacies were evaluated by contrast-enhanced CT performed every 6 weeks until 24 weeks after the initiation of both agents, and thereafter, every 9 weeks, according to the modified Response Evaluation Criteria in Solid Tumors (mRECIST) [15], in which tumor response is assessed as follows: CR: disappearance of any intratumor arterial enhancement in all target lesions; PR: at least a 30% decrease in the sum of the diameters of viable (enhancement in the arterial phase) target lesions; PD: an increase by at least 20% in the sum of the diameters of viable (enhancing) target lesions; SD: cases that did not qualify for either PR or PD. Adverse events were assessed according to the Common Terminology Criteria for Adverse Events (CTCAE) version 4.0 published by the National Cancer Institute [16].

## Statistical analysis

The $\chi^2$-test or Fisher's exact test was performed to compare the baseline characteristics of the patients and the therapeutic efficacies of atezolizumab plus bevacizumab treatment. A multivariate logistic regression analysis was performed to identify significant factors associated with the therapeutic efficacy. The Wilcoxon signed-rank test was used to compare the liver functions at the baseline and during the treatment. The PFS rates and cumulative OS rates after the initiation of atezolizumab plus bevacizumab therapy were calculated using the Kaplan-Meier method, and compared by the log-rank test. Factors associated with the survival rates were also analyzed using a Cox proportional hazard regression analysis model. *P* values of less than 0.05 were considered as denoting statistical significance.

## Results

### Demographic features and clinical characteristic of the patients

The demographic and clinical characteristics of the 82 patients (**Fig 1**) treated with atezolizumab plus bevacizumab are shown in **Table 1**. The patients consisted of 70 (85.4%) men and 12 (14.6%) women, with a median age of 73 years (range, 49 to 86 years). The baseline Child-Pugh scores were 5, 6, 7, and 8 in 40 (48.8%), 35 (46.7%), 5 (6.1%), and 2 (2.5%) patients, and

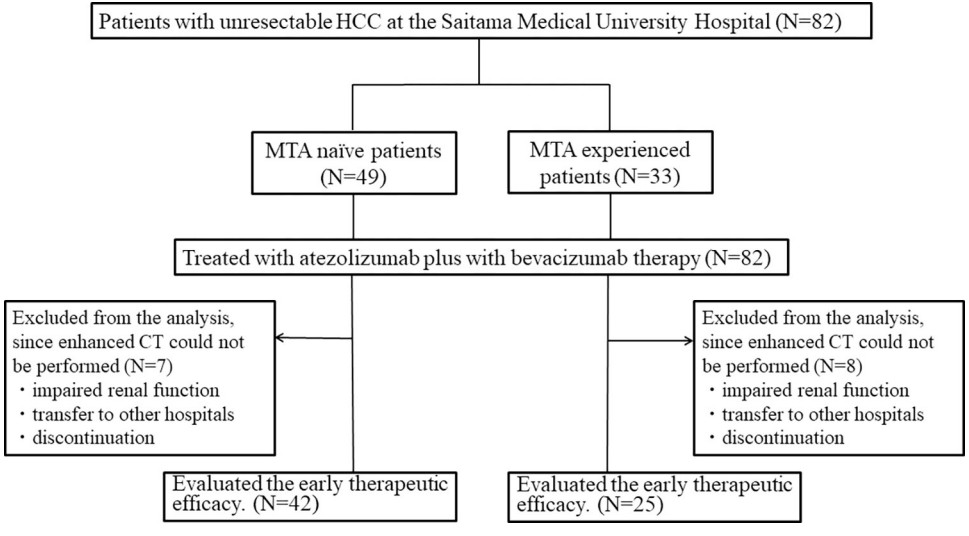

**Fig 1. Flow chart of the study population.**

**Table 1. Demographic features and clinical characteristics of the 82 hepatocellular carcinoma patients treated with atezolizumab plus bevacizumab.**

|  | Total (n = 82) | MTA-naïve patients (n = 49) | MTA-experienced patients (n = 33) | *P* value |
|---|---|---|---|---|
| Age: years old* | 73 (49–86) | 73 (49–86) | 73 (53–84) | 0.8574 |
| Sex: man / woman | 70 / 12 | 42 / 7 | 28 / 5 | >0.9999 |
| Etiology: HCV / HBV / alcohol / others | 24 / 7 / 21 / 30 | 14 / 3 / 12 / 20 | 10 / 4 / 9 / 10 | 0.4960 |
| Child-Pugh score: 5 / 6 / 7 / 8 | 40 / 35 / 5 / 2 | 28 / 18 / 2 / 1 | 12 / 17 / 3 / 1 | 0.3024 |
| ALBI grade: 1 / 2a / 2b | 29 / 17 / 36 | 21 / 12 / 16 | 8 / 5 / 20 | 0.0426 |
| BCLC stage: A / B / C | 3 / 31 / 48 | 1 / 13 / 35 | 2 / 18 / 13 | 0.0148 |
| Maximum diameter of tumors: mm* | 53 (10–180) | 69 (10–180) | 46 (11–94) | 0.0110 |
| Number of tumors: 1 / 2 / 3–9 / ≥10 | 17 / 7 / 33 / 25 | 12 / 4 / 20 / 13 | 5 / 3 / 13 / 12 | 0.6858 |
| Portal vein tumor thrombosis: Vp0 / 1 / 2 / 3 / 4 | 44 / 7 / 11 / 11 / 9 | 19 / 6 / 9 / 8 /7 | 25 / 1 / 2 / 3 / 2 | 0.0014 |
| Extrahepatic metastasis: absent / present | 54 / 28 | 30 / 19 | 24 / 9 | 0.3458 |
| AFP: < 200 ng/mL / ≥ 200 ng/mL | 40 / 42 | 23 / 26 | 17 / 16 | 0.8222 |
| N/L ratio: < 2.7 / ≥ 2.7 | 37 / 45 | 25 / 24 | 12 / 21 | 0.2584 |
| Previous liver resection: absent / present | 64 / 18 | 40 / 9 | 24 / 9 | 0.4177 |
| Previous RFA: absent / present | 71 / 11 | 45 / 4 | 26 / 7 | 0.1079 |
| Previous TACE or TAI: absent / present | 37 / 45 | 34 / 15 | 3 / 30 | <0.0001 |
| Radiation: absent / present | 77 / 5 | 48 / 1 | 29 / 4 | 0.1521 |

* Medium (range).

MTA: Molecular targeted agent, HCV: Hepatitis C virus, HBV: Hepatitis B virus, ALBI: Albumin bilirubin, BCLC: Barcelona Clinic Liver Cancer, AFP: Alpha-fetoprotein, N/L ratio: Neutrophil lymphocyte ratio, RFA: Radiofrequency ablation, TACE: Transcatheter arterial chemoembolization, TAI: Transcatheter arterial infusion chemotherapy.

the baseline mALBI grades were 1, 2a, and 2b in 29 (35.4%), 17 (20.7%), and 36 (43.8%) patients, respectively. The extents of HCC progression at the initiation of atezolizumab plus bevacizumab treatment were A, B, and C in 3 (3.7%), 31 (37.8%), and 48 (58.5%) patients, respectively. Tumor thrombosis in the portal vein was present in 38 patients (46.3%) and the extents were classified as Vp1 (subsegmentary), Vp2 (secondary-order branch), Vp3 (first-order branch), and Vp4 (main trunk) in 7, 11, 11, and 9 patients, respectively. Extrahepatic metastasis was present in 28 patients (34.1%).

Of the 82 patients, 33 (40.2%) had a previous history of treatment with MTAs: 29 patients had received lenvatinib, 1 patient had received sorafenib, 1 patient had received lenvatinib followed by ramucirumab, 1 patient had received lenvatinib followed by cabozantinib, and 1 patient had received lenvatinib followed by sorafenib and then regorafenib. Of the 29 patients who had received treatment with lenvatinib alone, the therapy was discontinued due to HCC progression and emergence of adverse events in 18 and 11 patients, respectively. In the remaining 4 patients who received other MTAs, the final therapy was discontinued due to HCC progression. Moreover, 18 (22.0%), 11 (13.4%), 45 (54.9%), and 5 (6.1%) patients had previously undergone liver resection, radiofrequency ablation (RFA), TACE/TAI, and radiation therapy, respectively. Consequently, 22 patients (26.8%) had previously undergone local therapies for HCC, including liver resection, RFA, TACE/TAI, or radiation, without systemic MTA treatment, 31 patients (37.8%) had received both local and systemic therapies, and 2 patients (2.4%) had received systemic therapies without local therapies, whereas 27 patients (32.9%) had not previously undergone either local or systemic therapy for HCC.

When the baseline demographic characteristics and clinical features were compared between MTA-naïve patients and MTA-experienced patients, the severity of liver damage

assessed by the mALBI grade, but not by the Child-Pugh scores was greater in the MTA-experienced patients than in the MTA-naïve patients (P = 0.0426). In contrast, HCCs were more advanced in the MTA-naive patients than in the MTA-experienced patients; the maximal tumor diameter was higher in the MTA-naïve patients than in the MTA-experienced patients (69 mm vs. 46 mm, P = 0.0110) and the number of HCC patients with complicating portal vein tumor thrombosis (PVTT) was higher in the MTA-naïve patients than in the MTA-experienced patients (61.2% vs. 24.2%, P = 0.0014). Consequently, percentage of patients with BCLC stage C HCC was higher in the MTA-naive group than in the MTA-experienced group (71.4% vs. 39.4%, P = 0.0148), even though the percentage of HCC patients with complicating extrahepatic metastasis was similar between the two groups (38.8% vs. 27.3%, P = 0.3458). The percentage of patients with a previous history of TACE and/or TAI was higher in the MTA-experienced group than that in the MTA-naïve group (30.6% vs. 90.9%, P<0.0001), while the percentages of patients who had undergone liver resection and radiation therapy were not significantly different between the two groups (18.4% vs. 27.3%, P = 0.4177; 2.1% vs. 12.1%, P = 0.1521).

## Early therapeutic efficacy of atezolizumab plus bevacizumab

Of the 82 patients who were treated with atezolizumab plus bevacizumab, 15 were excluded from the analysis of the early therapeutic efficacy, since contrast-enhanced CT examinations could not be performed in these patients due to impaired renal function, transfer to other hospitals, and/or discontinuation. Thus, the early therapeutic efficacy after 6 or 12 weeks of treatment was evaluated in a total of 67 patients. Of the 65 patients in whom the efficacy at 6 weeks was evaluated, none showed CR, while 18 (27.7%), 32 (49.2%), and 15 (23.1%) patients showed PR, SD, and PD, respectively. The objective response rate (ORR) and disease control rate (DCR) were calculated as 27.7% and 76.9%, respectively (**Fig 2A**). Thereafter, the efficacy at 12 weeks was evaluated in 43 patients, when 2 (4.7%), 8 (18.6%), 26 (60.5%), and 7 (16.3%) patients showed CR, PR, SD, and PD, respectively; the ORR and DCR at this time-point were 23.3% and 83.7%, respectively (**Fig 2B**). Thus, the best responses after 6 and 12 weeks of treatment evaluated in a total of 67 patients were CR in 2 patients (3.0%), PR in 20 patients (29.9%), SD in 33 patients (49.3%), and PD in 12 patients (17.9%), representing an ORR of 32.8% and DCR of 82.1%.

The early therapeutic efficacies evaluated after both 6 and 12 weeks of treatment were similar between patients with BCLC stage A/B HCC and BCLC stage C HCC (**Fig 2A and 2B**), and the ORR and DCR at best response were 33.3% and 85.2%, respectively, in the 27 patients with BCLC stage A/B HCC, and 32.5% and 80.0%, respectively, in the 40 patients with BCLC stage C HCC (P>0.9999 and P = 0.7490, respectively). The early therapeutic efficacies evaluated after 6 and 12 weeks of treatment were more favorable in the MTA-naïve patients than in the MTA-experienced patients (**Fig 2C and 2D**); the ORR at best response in the 42 MTA-naïve patients (40.5%) tended to be higher than that in the 25 MTA-experienced patients (20.0%) (P = 0.0948), while it was significantly higher than that in the 13 patients who received atezolizumab plus bevacizumab treatment immediately following discontinuation of previous lenvatinib therapy due to PD (7.7%) (P = 0.0410). The ORR at best response did not differ in patients with neutrophil lymphocyte ratio (N/L ratio) in the peripheral blood of less than 2.7 as compared to patients with the ratio of 2.7 or more (**Table 2A and 2B**). As shown in **Table 2B**, multivariate analysis revealed that a previous history of MTA therapy with lenvatinib as the sole factor tended to be associated with an unfavorable efficacy, with a HR of 4.846 (P = 0.0619). Thus, the significance of the N/L ratio was evaluated both in the MTA-naïve patients and MTA-experienced patients (**Table 3**), but he ORR at best response did not differ

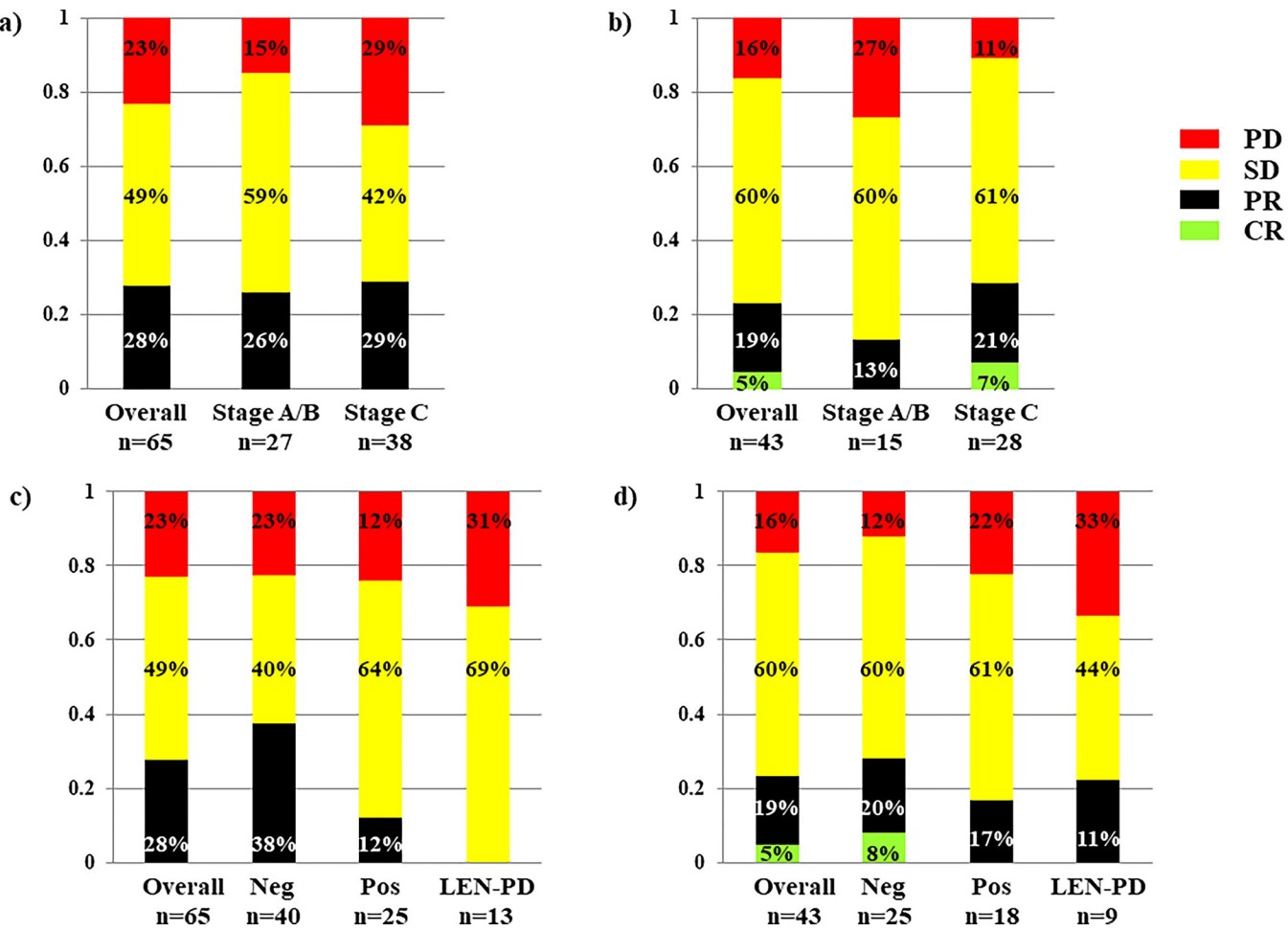

**Fig 2. Early therapeutic efficacy of atezolizumab (ATZ) plus bevacizumab (BEV) in patients with hepatocellular carcinoma assessed by the modified response evaluation criteria in solid tumors (mRECIST).** a) and c): Evaluation after 6 weeks of the therapy, b) and d): Evaluation after 12 weeks of the therapy. a) and b): Evaluation depending on the extent of HCC progression at the baseline according to the Barcelona Clinic Liver Cancer (BCLC) staging system. c) and d): Evaluation depending on the presence/absence of previous history of treatment with molecular-targeted agents (MTAs); neg and pos denote patients without and with a previous history of MTA treatment, respectively. LEN-PD denotes discontinuation of lenvatinib due to PD immediately prior to the start of ATZ plus BEV treatment.

between patients with the N/L ratio of less than 2.7 and those with the ratio of 2.7 or more in both groups.

## Outcomes of the patients treated with atezolizumab plus bevacizumab

During the medium follow-up period of 272 days (range; 14–699 days), atezolizumab plus bevacizumab was discontinued in 57 patients (69.5%) (due to progression of HCC in 32 patients, including 16 and 16 patients, respectively, and emergence of adverse events and/or deterioration of the performance status (PS) in 25 patients, including 16 and 9 patients, respectively, of the MTA-naïve and MTA-experienced groups). The PFS was evaluated in a total of 71 patients, including 67 patients who were included in the analysis of the early therapeutic efficacy and 4 patients who died before evaluation of the efficacy. As shown in Fig 3, the medium PFS was 169 days, and the PFS period tended to be longer in the MTA-naïve group (215 days) than in the MTA-experienced group (137 days) (P = 0.06).

**Table 2. Factors associated with the therapeutic efficacy of atezolizumab plus bevacizumab in patients with unresectable hepatocellular carcinoma (HCC) assessed based on the best response on contrast-enhanced CT according to the modified response evaluation criteria in solid tumors (mRECIST) after 6 and 12 weeks of treatment.**

| a) Evaluation in a total of 67 patients | | | | | | | | | | |
|---|---|---|---|---|---|---|---|---|---|---|
| | | Univariate analysis | | | | | Multivariate analysis | | | |
| | | Number of patients (%) | | | | p-values | Hazard ratio | 95% CI | p-values | |
| | Total | CR | PR | SD | PD | | | | | |
| Age: years | < 70 | 22 | 1 (4.5) | 07 (31.8) | 12 (54.5) | 02 (09.1) | 0.6675 | 1 | 0.319–3.420 | 0.9431 |
| | ≥ 70 | 45 | 1 (2.2) | 13 (28.9) | 21 (46.7) | 10 (22.2) | | 1.044 | | |
| Etiology | Viral | 23 | 1 (4.3) | 08 (34.8) | 10 (43.5) | 04 (17.4) | 0.4290 | 1 | 0.460–4.714 | 0.514 |
| | non-viral | 44 | 1 (2.3) | 12 (27.3) | 23 (52.3) | 08 (18.2) | | 1.473 | | |
| mALBI grades | 1/ 2a | 42 | 1 (2.4) | 12 (28.6) | 21 (50.0) | 08 (19.0) | 0.6707 | 1 | 0.191–2.089 | 0.4512 |
| | 2b | 25 | 1 (4.0) | 08 (32.0) | 12 (48.0) | 04 (16.0) | | 0.631 | | |
| BCLC Stages | A/ B | 27 | 0 (0.0) | 09 (33.3) | 14 (51.9) | 04 (14.8) | 0.9432 | 1 | 0.134–8.651 | 0.9456 |
| | C | 40 | 2 (5.0) | 11 (27.5) | 19 (47.5) | 08 (20.0) | | 1.075 | | |
| PVTT | Absent | 34 | 0 (0.0) | 11 (32.4) | 17 (50.0) | 06 (17.6) | 0.9319 | 1 | 0.148–9.219 | 0.8833 |
| | Present | 33 | 2 (6.1) | 09 (27.3) | 16 (48.5) | 06 (18.2) | | 1.167 | | |
| EH Metastasis | Absent | 45 | 1 (2.2) | 15 (33.3) | 22 (48.9) | 07 (15.6) | 0.4990 | 1 | 0.304–5.366 | 0.7383 |
| | Present | 22 | 1 (4.5) | 05 (22.7) | 11 (50.0) | 05 (22.7) | | 1.277 | | |
| AFP: ng/mL | < 200 | 32 | 0 (0.0) | 11 (34.4) | 18 (56.3) | 03 (09.4) | 0.7976 | 1 | 0.325–4.389 | 0.789 |
| | ≥ 200 | 35 | 2 (5.7) | 09 (25.7) | 15 (42.9) | 09 (25.7) | | 1.194 | | |
| N/L ratio | <2.7 | 32 | 0 (0.0) | 11 (34.4) | 13 (40.6) | 08 (25.0) | 0.7976 | 1 | 0.320–3.068 | 0.9872 |
| | ≥2.7 | 35 | 2 (5.7) | 09 (25.7) | 20 (57.1) | 04 (11.4) | | 0.991 | | |
| Previous TACE/TAI | Absent | 32 | 1 (3.1) | 11 (34.4) | 13 (40.6) | 7 (21.9) | 0.438 | 1 | 0.293–3.916 | 0.9176 |
| | Present | 35 | 1 (2.9) | 9 (25.7) | 20 (57.1) | 5 (14.3) | | 1.071 | | |
| MTA | Naïve | 42 | 2 (4.8) | 15 (35.7) | 18 (42.9) | 07 (16.7) | 0.0902 | 1 | 0.763–15.033 | 0.1087 |
| | Experienced | 25 | 0 (0.0) | 05 (20.0) | 15 (60.0) | 05 (20.0) | | 3.386 | | |
| b) Evaluation in 65 patients consisting of 42 MTA-naïve patients and 23 patients after lenvatinib discontinuation due to PD or adverse events | | | | | | | | | | |
| | | Univariate analysis | | | | | Multivariate analysis | | | |
| | | Number of patients (%) | | | | p-values | Hazard ratio | 95% CI | p-values | |
| | Total | CR | PR | SD | PD | | | | | |

*(Continued)*

**Table 2.** (Continued)

| | | | | | | | | | | |
|---|---|---|---|---|---|---|---|---|---|---|
| Age: years | < 70 | 22 | 1 (4.5) | 07 (31.8) | 12 (54.5) | 02 (09.1) | 0.6174 | 1 | 0.273–3.233 | 0.9201 |
| | ≥ 70 | 43 | 1 (2.3) | 12 (27.9) | 20 (46.5) | 10 (23.3) | | 0.939 | | |
| Etiology | Viral | 22 | 1 (4.5) | 08 (36.4) | 09 (40.9) | 04 (18.2) | 0.2914 | 1 | 0.508–5.702 | 0.3883 |
| | non-viral | 43 | 1 (2.3) | 11 (25.6) | 23 (53.5) | 08 (18.6) | | 1.702 | | |
| mALBI grades | 1/ 2a | 41 | 1 (2.4) | 11 (26.8) | 21 (51.2) | 08 (19.5) | 0.4943 | 1 | 0.142–1.760 | 0.2801 |
| | 2b | 24 | 1 (4.2) | 08 (33.3) | 11 (45.8) | 04 (16.7) | | 0.499 | | |
| BCLC Stages | A/ B | 27 | 0 (0.0) | 09 (33.3) | 14 (51.9) | 04 (14.8) | 0.8815 | 1 | 0.220–22.011 | 0.502 |
| | C | 38 | 2 (5.3) | 10 (26.3) | 18 (47.4) | 08 (21.1) | | 2.201 | | |
| PVTT | Absent | 33 | 0 (0.0) | 10 (30.3) | 17 (51.5) | 06 (18.2) | 0.7257 | 1 | 0.073–6.288 | 0.7305 |
| | Present | 32 | 2 (6.3) | 09 (28.1) | 15 (46.9) | 06 (18.8) | | 0.676 | | |
| EH Metastasis | Absent | 45 | 1 (2.2) | 15 (33.3) | 22 (48.9) | 07 (15.6) | 0.4034 | 1 | 0.274–5.645 | 0.7774 |
| | Present | 20 | 1 (5.0) | 04 (20.0) | 10 (50.0) | 05 (25.0) | | 1.244 | | |
| AFP: ng/mL | < 200 | 31 | 0 (0.0) | 10 (32.3) | 18 (58.1) | 03 (09.7) | 0.9935 | 1 | 0.316–4.485 | 0.7972 |
| | ≥ 200 | 34 | 2 (5.9) | 09 (26.5) | 14 (41.2) | 09 (26.5) | | 1.19 | | |
| N/L ratio | <2.7 | 31 | 0 (0.0) | 10 (32.3) | 13 (41.9) | 08 (25.8) | 0.9935 | 1 | 0.239–2.577 | 0.6889 |
| | ≥2.7 | 34 | 2 (5.9) | 09 (26.5) | 19 (55.9) | 04 (11.8) | | 0.784 | | |
| Previous TACE/TAI | Absent | 32 | 1 (3.1) | 11 (34.4) | 13 (40.6) | 7 (21.9) | 0.3796 | 1 | 0.296–4.321 | 0.8563 |
| | Present | 33 | 1 (3.9) | 8 (24.2) | 19 (57.6) | 5 (15.2) | | 1.132 | | |
| Lenvatinib | Naïve | 42 | 2 (4.8) | 15 (35.7) | 18 (42.9) | 07 (16.7) | 0.0642 | 1 | 0.924–25.409 | 0.0619 |
| | experienced | 23 | 0 (0.0) | 04 (17.4) | 14 (60.9) | 05 (21.7) | | 4.846 | | |

**Table 3. The relation of neutrophil lymphocyte ratio (N/L ratio) in the peripheral blood and therapeutic efficacy of atezolizumab plus bevacizumab in patients with unresectable hepatocellular carcinoma (HCC) assessed based on the best response on contrast-enhanced CT according to the modified response evaluation criteria in solid tumors (mRECIST) at the best response.**

| | | MTA-naïve patients | | | | | MTA-experienced patients | | | | |
|---|---|---|---|---|---|---|---|---|---|---|---|
| | | (n = 42) | | | | | (n = 25) | | | | |
| | | CR | PR | SD | PD | *P* Values | CR | PR | SD | PD | P values |
| N/L ratio | <2.7 | 0 | 9 | 7 | 500 | >0. 9999 | 0 | 2 | 6 | 30 | >0.9999 |
| | ≥2.7 | 2 | 6 | 11 | 2 | | 0 | 3 | 9 | 2 | |

MTA: Molecular targeted agent, N/L ratio: Neutrophil lymphocyte ratio.

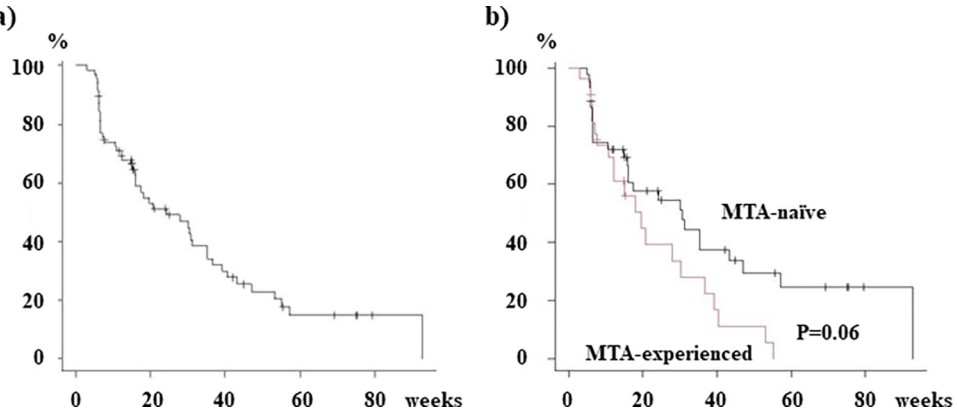

**Fig 3. Progression-free survival (PFS) of patients treated with atezolizumab (ATZ) plus bevacizumab (BEV).** a) PFS in a total of 71 patients. b) PFS depending on the presence/absence of previous history of chemotherapy with molecular-targeted agents (MTAs); 44 MTA-patients and 27 MTA-experienced patients.

Of the 57 patients in whom atezolizumab plus bevacizumab was discontinued, 26 patients received additional therapies for HCC; the percentage of patients who received additional therapies relative to those in whom atezolizumab plus bevacizumab was discontinued were similar between the MTA-naïve (43.8%; 14/32) and MTA-experienced (48.0%; 12/25) groups. As second-line therapies, 2 patients received TACE and transcatheter arterial embolization (TAE) and 24 patients received treatment with MTAs (lenvatinib: 20; cabozantinib: 2; ramucirumab: 2). As third- or later-line therapies, 3 patients received TACE/TAE, 7 patients received systemic therapy with MTAs (lenvatinib: 3, cabozantinib: 3, ramucirumab: 1) and 4 patients received atezolizumab plus bevacizumab therapy.

Evaluation of the OS in the total subject population of 82 patients (**Fig 4**) revealed OS rates of 85.8% and 71.8% at 24 weeks and 48 weeks, respectively, after the initiation of atezolizumab plus bevacizumab therapy, and the rates were not significantly different between the MTA naïve patients (84.9% and 70.6%, respectively) and MTA-experienced patients (87.6% and 73.6%, respectively). Multivariate analysis identified the peripheral N/L ratio and early therapeutic efficacy as factors associated with the OS rates (**Table 4**). The HRs were 0.351 for patients with N/L ratios of less than 2.7 as compared to patients with N/L ratios of 2.7 or more (P = 0.0303), and 0.124 in the patients who showed favorable treatment responses (CR, PR, SD or indeterminate status in whom atezolizumab plus bevacizumab was continued for longer than 12 weeks) as compared with that in the patients who failed to show favorable treatment responses (PD or indeterminate status, in whom atezolizumab plus bevacizumab therapy was discontinued within 12 weeks) (P<0.0001).

## Adverse events

As shown in **Table 5**, some or the other adverse event(s) was seen in 77 patients (93.9%); the adverse events were grade 1, grade 2, grade 3 and grade 4 in 14 (17.1%), 41 (50.0%), 20 (24.4%) and 2 (2.4%) patients, respectively. Immune-related adverse events (irAEs) developed in 2 patients (2.4%: colitis in 1 patient and endocrinological disorders in 1 patient), of whom one required glucocorticoid therapy for resolution of the adverse event. Proteinuria and decrease in serum albumin levels were seen in 64 patients (78.0%) and 39 patients (47.6%), respectively. Consequently, atezolizumab plus bevacizumab therapy had to be discontinued in 20 patients (24.4%) due to the occurrence of adverse events. Proteinuria and decrease in serum albumin levels were seen in 64 patients (78.0%) and 39 patients (47.6%), respectively. Consequently,

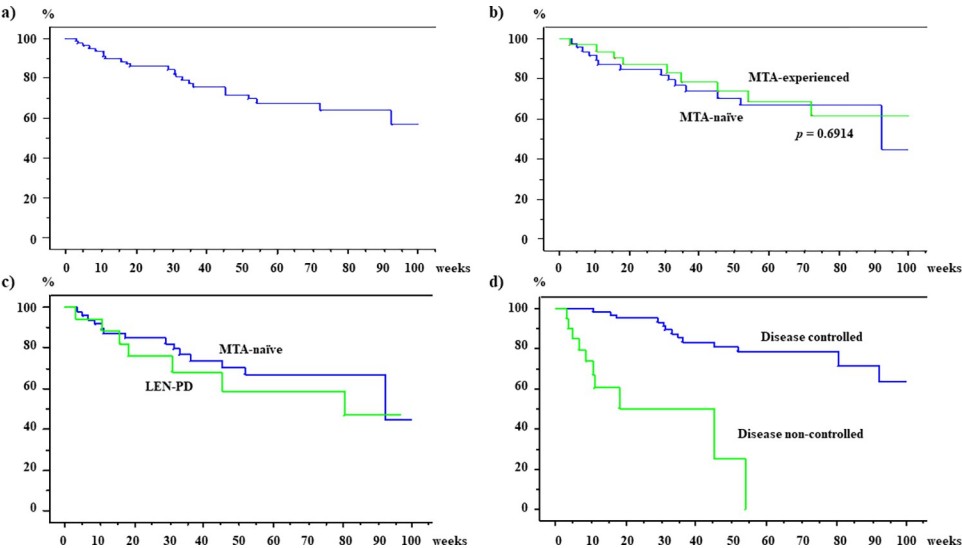

**Fig 4. Overall survival (OS) of hepatocellular carcinoma patients treated with atezolizumab (ATZ) Plus bevacizumab (BEV).** a) OS in a total of 82 patients. b) OS depending on the presence/absence of previous history of treatment with molecular-targeted agents (MTAs); 49 MTA-naïve patients and 33 MTA-experienced patients. c) OS in patients who began to receive ATZ plus BEV immediately following discontinuation of lenvatinib (LEN) due to PD. d) OS depending on the disease control status; 61 patients who were assessed as showing CR, PR, SD or indeterminate status (disease control) in whom ATZ plus BEV was continued for longer than 12 weeks, and 21 patients assessed as showing PD or indeterminate status in whom ATZ plus BEV was discontinued within12 weeks of the start of therapy.

atezolizumab plus bevacizumab therapy had to be discontinued in 20 patients (24.4%) due to the occurrence of adverse events.

Changes in the liver function were assessed by measurement of the ALBI score in 73 patients, after exclusion of 9 patients in whom atezolizumab plus bevacizumab was discontinued within 6 weeks of the start of therapy. Deterioration of the median ALBI score, as compared with the baseline (-2.42, range -3.61 to -1.45) was observed after both 6 weeks (-2.09, range -3.28 to -0.73) and 12 weeks (-2.19, range -3.16 to -0.96) of therapy (P = 0.0004 and P = 0.0005, respectively) (Fig 5A). Such deterioration was observed irrespective of whether the patients manifested mALBI grade 1, grade 2a, or grade 2b at the baseline (Fig 5B–5D) and irrespective of whether the patients showed no proteinuria or CTCAE grade 1 or grade 2 proteinuria within 12 weeks of initiation of atezolizumab plus atezolizumab treatment (Fig 5E–5G).

## Discussion

In the present study, the therapeutic efficacy of atezolizumab plus bevacizumab was evaluated in 82 patients with unresectable HCC, and the ORR and DCR based on the best responses after 6 and 12 weeks of therapy assessed according to the mRECIST on contrast-enhanced CT were 32.8% and 82.1%, respectively. Neither the ORR nor the DCR differed significantly between patients with the BCLC stage A/B HCC and those with BCLC stage C HCC. The ORR in the present study was almost similar to that reported from previous studies: 33.2% in the IMbrave150 trial [3], and 22.5% by Ando et al. [17], 28.7% by Chuma et al. [12], 29.0% by Himmelsbach et al. [18], 30.4% by Kuzuya et al. [19] and 43.8% by Maesaka et al. [20] according to the mRECIST in real-world practice. Previously, we reported the therapeutic efficacy of lenvatinib in 69 patients with unresectable HCC: the ORR and DCR were 48.1% and 85.2%, respectively, as assessed by contrast-enhanced CT according to the mRECIST, and the ORR

**Table 4. Factors associated with the aumulative survival rates of 82 patients with unresectable hepatocellular carcinoma (HCC) receiving atezolizumab (ATZ) plus bevacizumab (BEV) rreatment.**

| | | Kaplan Meier | | | | Cox proportional hazard regression | | |
|---|---|---|---|---|---|---|---|---|
| | | survival rates (%) | | | p-values | Hazard ratio | 95%CI | p-values |
| | | Total | 24w | 48w | | | | |
| Age: years | < 70 | 27 | 84.1 | 73.5 | 0.6852 | | | |
| | ≥ 70 | 55 | 86.8 | 70.8 | | | | |
| Etiology | Viral | 31 | 93.3 | 71.6 | 0.3648 | | | |
| | non-viral | 51 | 81.6 | 73.1 | | | | |
| mALBI grade | 1/ 2a | 46 | 95.6 | 78.7 | 0.3371 | | | |
| | 2b | 36 | 73.3 | 63.2 | | | | |
| BCLC stages | A/B | 34 | 91 | 83.1 | 0.0684 | | | |
| | C | 48 | 82.1 | 63.8 | | | | |
| PVTT | Absent | 44 | 88.4 | 79.3 | 0.1986 | | | |
| | Present | 38 | 82.8 | 62.7 | | | | |
| EH | Absent | 54 | 88.4 | 80.3 | 0.0312 | | | |
| Metastasis | Present | 28 | 80.5 | 55.2 | | | | |
| AFP: ng/mL | < 200 | 40 | 87.4 | 80.8 | 0.1209 | | | |
| | ≥ 200 | 42 | 83.8 | 61.7 | | | | |
| N/L ratio | < 2.7 | 37 | 91 | 83.2 | 0.0228 | 0.351 | 0.136–0.905 | 0.0303 |
| | ≥ 2.7 | 45 | 81.7 | 62.3 | | 1 | | |
| Previous TACE/TAI | Absent | 37 | 79.6 | 68.2 | 0.8448 | | | |
| | Present | 45 | 90.7 | 71.2 | | | | |
| MTA | Naïve | 49 | 84.9 | 70.6 | 0.6663 | | | |
| | experienced | 33 | 87.2 | 73.6 | | | | |
| Therapeutic | CR+PR | 22 | 95.5 | 75.2 | 0.4577 | | | |
| Efficacy | SD+PD | 45 | 92.5 | 78.1 | | | | |
| | ID | 15 | 47.1 | 47.1 | | | | |
| | CR+PR+SD+ID[1] | 61 | 95.1 | 81 | < 0.0001 | 0.124 | 0.051–0.302 | < 0.0001 |
| | PD+ID[2] | 21 | 50.2 | 25.1 | | 1 | | |

CI: Confidential interval, mALBI: Modified albumin bilirubin, BCLC: Barcelona Clinic Liver Cancer, PVTT; portal vein tumor thrombosis, EH Metastasis: Extrahepatic metastasis, AFP: Alpha-fetoprotein, N/L ratio: Neutrophil lymphocyte ratio, MTA: Molecular targeted agent, CR: Complete response, PR: Partial response, SD: Stable disease, PD: Progressive disease, ID: Indeterminate, ID[1]: Indeterminate but ATZ plus BEV was continued later than 12 weeks of the therapy, ID[2]: Indeterminate and ATZ plus BEV was discontinued within12 weeks of the therapy.

was significantly higher in patients with the BCLC stage A/B HCC than in those with BCLC stage C HCC (67.7% vs 21.7%) [11]. The overall early therapeutic efficacy of atezolizumab plus bevacizumab was inferior to that of lenvatinib treatment in patients with the BCLC stage A/B HCC, since we previously reported the ORR and DCR during the chemotherapy with lenvatinib in patients with the BCLC stage A/B HCC were 67.7% and 90.3%, respectively. Maesaka *et. al.*, however, reported that no significant differences were found between patients receiving lenvatinib and those receiving atezolizumab plus bevacizumab in the ORR after propensity score matching especially in MTA-naïve patients [20]. Thus, the early therapeutic effect of atezolizumab plus bevacizumab should be further evaluated in future.

Atezolizumab plus bevacizumab is recommended as a first-line therapy for patients who are candidates for the systemic chemotherapy. However, atezolizumab plus bevacizumab has been used for a lot of patients with previous chemotherapy with MTAs in clinical practice. Thus, the significance of atezolizumab plus bevacizumab as a second-line therapy need to be

**Table 5. Adverse events seen during atezolizumab (ATZ) plus bevacizumab (BEV) in 82 patients with unre sectable hepatocellular carcinoma (HCC).**

| Events | Number of Patients (%) | | | | |
|---|---|---|---|---|---|
| | Total | Grading: CTCAE version 4 | | | |
| | | 1 | 2 | 3 | 4 |
| Any adverse events | 77 (94) | 14 (17) | 41 (50) | 20 (24) | 2 ( 2) |
| Rash | 08 (10) | 05 (06) | 03 (04) | 0 | 0 |
| Diarrhea | 03 (04) | 02 (02) | 01 (01) | 0 | 0 |
| Colitis | 01 (01) | 0 | 01 (01) | 0 | 0 |
| decreased appetite and/or nausea | 21 (26) | 14 (17) | 04 (05) | 03 (04) | 0 |
| general fatigue | 17 (21) | 16 (20) | 0 | 01 (01) | 0 |
| Ascites | 11 (13) | 01 (01) | 10 (12) | 0 | 0 |
| Hypertension | 03 (04) | 0 | 02 (02) | 01 (01) | 0 |
| hepatic encephalopathy | 01 (01) | 0 | 01 (01) | 0 | 0 |
| Leukopenia | 10 (12) | 2 (2) | 7 (9) | 01 (01) | 0 |
| Anemia | 26 (32) | 14 (17) | 07 (09) | 05 (06) | 0 |
| Thrombocytopenia | 16 (20) | 11 (13) | 02 (02) | 03 (04) | 0 |
| increased serum total bilirubin | 22 (27) | 10 (12) | 08 (10) | 03 (04) | 01 (01) |
| elevated serum alanine aminotransferase | 27 (33) | 19 (23) | 07 (09) | 01 (01) | 0 |
| decreased serum albumin | 39 (48) | 06 (07) | 26 (32) | 07 (09) | 0 |
| increased serum creatinine | 19 (23) | 15 (18) | 03 (04) | 01 (01) | 0 |
| Hypothyroidism | 01 (01) | 0 | 01 (01) | 0 | 0 |
| Proteinuria | 64 (78) | 42 (51) | 16 (20) | 06 (07) | 0 |
| adrenal insufficiency | 01 (01) | 0 | 0 | 01 (01) | 0 |
| infusion reaction | 01 (01) | 0 | 0 | 0 | 01 (01) |
| GI bleeding | 3 (04) | 0 | 0 | 03 (04) | 0 |

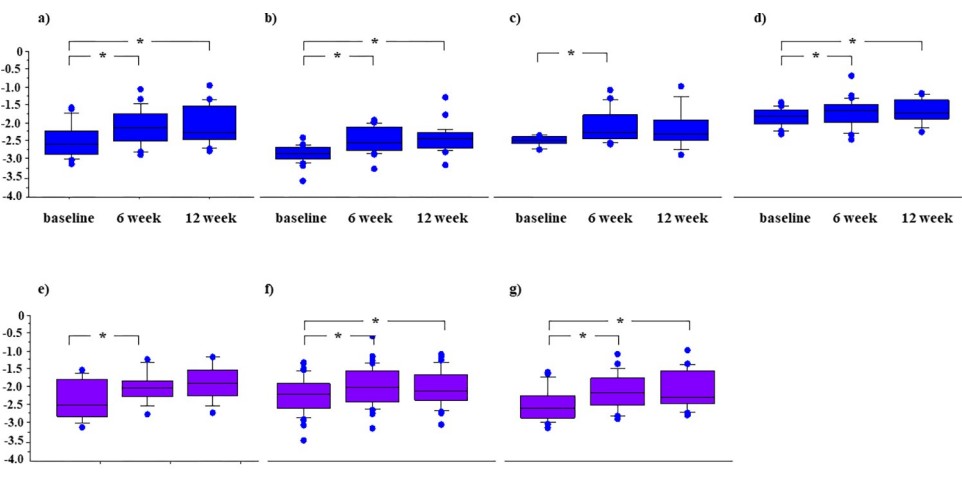

**Fig 5. Albumin-bilirubin (ALBI) scores in hepatocellular carcinoma patients treated with atezolizumab (ATZ) plus bevacizumab (BEV).** a) Score in a total of 73 patients, b) 28 patients with mALBI grade-1 at baseline, c) 16 patients with mALBI grade-2a at baseline, d) 29 patients with mALBI grade-2b at baseline, e) 13 patients without proteinuria within 12 weeks of initiation of ATZ plus BEV treatment. f) 39 patients with grade 1 proteinuria (according to the Common Terminology Criteria for Adverse Events (CTCAE)) within 12 weeks of initiation of ATZ plus BEV treatment, g) 21 patients with grade 2 proteinuria (CTCAE) within 12 weeks of initiation of ATZ plus BEV treatment.

elucidated. We assessed the factors associated with early therapeutic efficacy of atezolizumab plus bevacizumab, and multivariate analysis identified previous history of MTA therapy with lenvatinib as the sole factor tended to be associated with an unfavorable therapeutic efficacy. The ORR tended to be lower in the patients in whom atezolizumab plus bevacizumab was initiated following discontinuation of prior lenvatinib therapy due to PD or adverse events than in MTA-naïve patients. In the present study, 17 of the 42 MTA-naïve patients (40.5%) showed CR or PR, indicating that the ORR of these patients was almost similar to that reported by Maesaka *et al*. (43.8%) [20], who exclusively enrolled MTA-naïve patients in their study. Moreover, even according to previous reports, the ORR was higher in MTA-naïve patients than in MTA-experienced patients; 36.0% and 20.5%, respectively, as reported by Chuma *et al*. [12] and 37.5% and 12.5%, respectively, as reported by Ando *et al*. [17], although the difference was not significant in either report. Considering these observations, first-line chemotherapy with atezolizumab plus bevacizumab is recommended in patients with unresectable HCC, especially those with BCLC stage C HCC.

In the present study, the medium PFS period was 215 days in the MTA-naïve group and 137 days in the MTA-experienced group; the period in the MTA-naïve patients was similar to that reported from previous observations in the clinical trial and real-world practice: 196 days reported by Himmelsbach *et al*. [18], 6.8 months reported from the IMbrave150 trial [3], 6.9 months reported by Fulgenzi *et al*. [21], and 8.8 months reported by Maesaka *et al*. [20]. Moreover, the OS rates were also similar to previous observations: the rate at 6 months was 85.8% in the present study and 84.8% in the IMbrave150 trial [3]. Therefore, factors associated with the OS were evaluated, and multivariate analysis revealed that achievement of disease control, but not early therapeutic response, was a factor contributing to a favorable outcome. In the present study, patients in whom the early therapeutic efficacy was indeterminate, but the therapy was continued were later classified into those who showed disease control in addition to patients who showed CR, PR, and SD. Consequently, the OS rate was significantly higher in the patients who showed disease control than in those who failed to achieve disease control. Previously, we reported that the OS rate at 48 weeks after initiation of lenvatinib therapy was significantly higher among patients who showed CR or PR than among those who did not show CR or PR [11]. Achievement of CR or PR is essential for a favorable outcome in patients receiving lenvatinib therapy, while a favorable outcome can be expected following atezolizumab plus bevacizumab therapy even in patients showing SD, besides those showing the best treatment response of CR or PR. Moreover, multivariate analysis identified N/L ratio of less than 2.7 as a factor associated with a favorable outcome, despite that the ORR did not differ between patients with the ratio of less than 2.7 and those with the ration of 2,7 or more. The immunological backgrounds influencing the long-term outcomes of patients receiving atezolizumab plus bevacizumab should be investigated in the future.

In the present study, PFS and OS after initiation of atezolizumab plus bevacizumab in the MTA-experienced patients were compared to those in the MTA-naïve patients, and the medium PFS was 137 days, and OS rates at 48 weeks was 73.6% in the MTA-experienced patients. Although previous history of MTA therapy tended to be associated with unfavorable early therapeutic efficacy and PFS, the OS rates did not differ between MTA-naïve and MTA-experienced patients. As shown in **Table 1**, a percentage of patients with BCLC stage C was higher in the MTA-naïve patients than in the MTA-experienced patients. In contrast, a percentage of patients receiving additional therapies after discontinuation of atezolizumab plus bevacizumab were similar between both groups. These data suggest that the systemic therapy with atezolizumab plus bevacizumab merits consideration as a second-line therapy even in the MTA-experienced patients. Also, the most effective additional therapy after atezolizumab plus bevacizumab should be identified in the future.

In the present study, one or the other adverse event(s) was seen in almost all patients receiving treatment with atezolizumab plus bevacizumab. However, irAEs developed only in 2 patients, one of whom required glucocorticoid administration for resolution of the adverse event. In contrast, proteinuria and/or decrease of the serum albumin level was seen in almost about 80% of patients. Consequently, atezolizumab plus bevacizumab had to be discontinued due to the emergence of adverse events in 24% of the patients. Thus, management of adverse events, especially that of proteinuria, is crucial to improve the outcomes of patients receiving atezolizumab plus bevacizumab therapy. Of note, significant deterioration of the median ALBI scores, as compared with the baseline, was observed after both 6 and 12 weeks of therapy, and such deterioration was seen irrespective of whether the patients had no proteinuria, CTCAE grade 1 or grade 2 proteinuria within 12 weeks of initiation of atezolizumab plus bevacizumab treatment. Similar observations were also reported by Maesaka *et al.* [20]. Thus, the effects of atezolizumab plus bevacizumab therapy on the liver function needs to be more precisely investigated in the future.

The limitations in the present study were as follows. First, the study was conducted retrospectively at a single institute, in a small patient cohort. Second, the medium follow-up period in the present study was only 272 days. Although the early therapeutic efficacy and intermediate-term outcome of patients receiving atezolizumab plus bevacizumab were evaluated, the outcomes of the patients over the long term still need to be evaluated. Moreover, most of the patients received varied therapies for HCC following discontinuation of atezolizumab plus bevacizumab, including MTA and TACE/TAE treatments. Therefore, the effects of the heterogeneity of additional therapies after atezolizumab plus bevacizumab therapy on the intermediate-term outcomes of the patients need to be evaluated in a large cohort in the future.

In conclusion, the early therapeutic efficacy of atezolizumab plus bevacizumab was superior in MTA-naïve patients than in MTA-experienced patients. However, atezolizumab plus bevacizumab treatment still merits consideration, even in MTA-experienced patients, as the OS rates determined after longer periods of treatment (24 and 48 weeks) were similar between the MTA-naïve patients and MTA-experienced patients.

## Supporting information

**S1 Checklist. TREND statement checklist.**
(PDF)

**S1 File. Study protocol (original).**
(DOCX)

**S2 File. Study protocol (english).**
(DOC)

**S3 File. Data.**
(PDF)

## Author Contributions

**Writing – original draft:** Shinpei Yamaba.

**Writing – review & editing:** Yukinori Imai, Kayoko Sugawara, Yoshihito Uchida, Akira Fuchigami, Hiroshi Uchiya, Nobuaki Nakayama, Satoshi Mochida.

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
