## [Decision Letter · Decision Letter 0]

7 Dec 2023

PONE-D-23-25052Usefulness of Atezolizumab Plus Bevacizumab as Second-line Therapy for Patients with Unresectable Hepatocellular CarcinomaPLOS ONE

Dear Dr. MOCHIDA,

Thank you for submitting your manuscript to PLOS ONE. After careful consideration, we feel that it has merit but does not fully meet PLOS ONE’s publication criteria as it currently stands. Therefore, we invite you to submit a revised version of the manuscript that addresses the points raised during the review process.

We look forward to receiving your revised manuscript.

Kind regards,

Jincheng Wang

Academic Editor

PLOS ONE

Journal Requirements:

2. Please confirm whether the ethics committee approved the use of opt-out consent

**Additional Editor Comments:**

1. The reviewer raises significant concerns about the novelty of your study. It is imperative that you clearly differentiate your work from existing literature and articulate the unique contributions of your study, particularly in the context of patients treated with Atezolizumab and Bevacizumab who have experienced MTA.

2. The limited number of patients in your study has been noted as a significant limitation. I advise you to acknowledge this explicitly and discuss how this impacts the generalizability of your findings. If possible, provide a justification for the sample size and explain how your study still provides valuable insights despite this limitation.

3. Please make thorough discussions regarding the clinical impact of ATZ/BEV as second-line therapy, given its current recommendation as a first-line treatment.

4. There are concerns about the clarity and consistency of your data presentation, particularly in Table 2 and Figure 2d. These should be revised for accuracy and coherence.

5. The significant difference in patient backgrounds (history of TACE or TAI) between groups needs to be addressed. This could potentially impact the results and should be discussed.

6. Additional analysis is requested regarding the neutrophil/lymphocyte ratio in different patient groups (MTA-treated vs. untreated). This should be included to strengthen your findings.

Reviewers' comments:

Reviewer's Responses to Questions

**Comments to the Author**

1. Is the manuscript technically sound, and do the data support the conclusions?

Reviewer #1: Partly

Reviewer #2: Yes

Reviewer #3: Partly

2. Has the statistical analysis been performed appropriately and rigorously? 

Reviewer #1: No

Reviewer #2: Yes

Reviewer #3: Yes

3. Have the authors made all data underlying the findings in their manuscript fully available?

Reviewer #1: Yes

Reviewer #2: Yes

Reviewer #3: Yes

4. Is the manuscript presented in an intelligible fashion and written in standard English?

Reviewer #1: Yes

Reviewer #2: Yes

Reviewer #3: Yes

5. Review Comments to the Author

Reviewer #1: Authors retrospectively evaluated the treatment results in patients treated with Atez/Bev according to MTA experienced or not. As the results, authors concluded that Atez/Bev might be useful in patients with MTA experienced. Similar works have been already found a lot, and this work is therefore lacking in novelty. Furthermore, the number of patients included to present study was limited. Multivariate analyses were therefore lacking the significance.

Reviewer #2: Yamaba and colleagues assessed the effectiveness of ATZ/BEV as a second-line therapy following therapy with MTA. The manuscript is intriguing and well-written. The results of this study contribute valuable data to the existing literature in this field, but there are several significant concerns.

1. As pointed out by the author in the manuscript, the current guidelines recommend ATZ/BEV as the first-line therapy for unresectable HCC. So, when considering the utility of ATZ/BEV as a second-line therapy, which is the primary discovery and focal point of this study, its impact on clinical practice seems to be somewhat constrained. This should be thoroughly discussed in the Discussion section.

2. In the Discussion section, the authors assert, "Thus, the overall early therapeutic efficacy of atezolizumab plus bevacizumab was inferior to that of lenvatinib treatment, even though atezolizumab plus bevacizumab still merits consideration in patients with the BCLC stage C HCC." Is this statement entirely accurate? The superiority of the treatment effect of both should be mentioned with reference to actual direct comparative studies of LEN and ATZ/BEV in treatment-naïve patients.

3. Please discuss why there was a difference in early treatment response and PFS, while there was no difference in OS between MTA-naïve and MTA-experienced patients.

4. The analysis in Table 2 may appear somewhat verbose. It would be clear to narrow it down to whether the patients have a history of receiving lenvatinib and conduct the analysis accordingly.

5. The percentage of LEN-PD in Figure 2d appears to be incorrect. Also, in Table 2, there are 23 patients who started ATZ/BEV with lenvatinib PD, which is not consistent with the text or FIGURE 2, so please check. Otherwise, the Result section is cumbersome, as noted above. Please reorganize and improve it for clarity.

6. There is a mess of "ATZ plus BEV" and "atezolizumab plus bevacizumab" in the text. Please check carefully.

Reviewer #3: The authors compared the patients with previous therapies with molecular-targeted agents (MTA) and the MTA-naïve patients. They concluded that there was no difference in overall survival between the two groups, suggesting that ATZ/BEV therapy may be useful for MTA-experienced patients.

Major

1. The number of patients studied was 82 in total, 33 of whom had been previously treated with MTA and 49 of whom had not been previously treated with MTA. Although the number of cases is large from a single institution, the number of cases is small, and the power of analysis is low to conclude that there is no difference by comparison. In addition, this is a retrospective study, so it is difficult to obtain a definitive conclusion from this study design.

2. In the comparison of patient backgrounds between the MTA-experienced and MTA-naïve groups, there is a significant difference in the history of TACE or TAI. This could be a major difference in patient background, and the impact of this difference on the results of the analysis cannot be ignored.

3. Although there is a difference in early treatment effect between the MTA-treated and untreated groups, it is necessary to explain why there is no difference in OS in those groups. The multivariate analysis shows that early treatment effect is associated with OS, but this contradicts the conclusion that there is no difference in OS between the two groups.

4. The achievement of CR, PR, or SD is associated with the neutrophil/lymphocyte count ratio in the peripheral blood. Should the neutrophil/lymphocyte count ratio in hepatocellular carcinoma tissue be the same as the ratio in peripheral blood? Also, it is unclear why immune checkpoint inhibitors are more effective when neutrophils outnumber lymphocytes.

5. Please provide the results of the analysis of the relationship between the achievement of CR, PR, or SD and the neutrophil/lymphocyte ratio in the peripheral blood, dividing the patients into two groups: those who have been treated with MTA and those who have not been treated with MTA.

6. PLOS authors have the option to publish the peer review history of their article (what does this mean?). If published, this will include your full peer review and any attached files.

Reviewer #1: No

Reviewer #2: No

Reviewer #3: No

---

## [Author Response · Author response to Decision Letter 0]

6 Jan 2024

Enclosed please find our revised manuscript entitled “Usefulness of Atezolizumab Plus Bevacizumab as Second-line Therapy for Patients with Unresectable Hepatocellular Carcinoma” for publication as an original article in PLOS ONE. A point-by-point response to the comments by the reviewers is as follows.

To Reviewer-2

1. As pointed out by the author in the manuscript, the current guidelines recommend ATZ/BEV as the first-line therapy for unresectable HCC. So, when considering the utility of ATZ/BEV as a second-line therapy, which is the primary discovery and focal point of this study, its impact on clinical practice seems to be somewhat constrained. This should be thoroughly discussed in the Discussion section.

We agree with the comment by the reviewer. Currently, ATZ/BEV is recommended as a first-line therapy for patients who are candidates for the systemic chemotherapy. However, ATZ/BEV has been used for a lot of patients with a previous treatment with MTAs in clinical practice. Thus, the significance of ATZ/BEV as a second-line therapy need to be elucidated. We added these descriptions and the significance of ATZ/BEV as a second-line chemotherapy in Discussion (P14, L26-L29). 

2. In the Discussion section, the authors assert, "Thus, the overall early therapeutic efficacy of atezolizumab plus bevacizumab was inferior to that of lenvatinib treatment, even though atezolizumab plus bevacizumab still merits consideration in patients with the BCLC stage C HCC." Is this statement entirely accurate? The superiority of the treatment effect of both should be mentioned with reference to actual direct comparative studies of LEN and ATZ/BEV in treatment-naïve patients.

We agree with the comment by the reviewer. The superiority of therapeutic effect should be mentioned with reference to actual direct comparative studies of lenvatinib and ATZ/BEV in treatment-naïve patients. Maesaka et.al reported that no significant differences were noted between two treatment groups in term of objective response rates after propensity score matching. We cited this manuscript and revised the description about this comparative study in Discussion (P14, L19-L25).

3. Please discuss why there was a difference in early treatment response and PFS, while there was no difference in OS between MTA-naïve and MTA-experienced patients.

As pointed out by the reviewer, the OS rates did not differ between MTA-naïve and MTA-experienced patients, while previous history of MTA therapy tended to be associated with unfavorable early therapeutic efficacy and PFS. As shown in Table 1, a percentage of patients with BCLC stage C was higher in the MTA- naïve patients than in the MTA-experienced patients. In contrast, a percentage of patients receiving additional therapies after discontinuation of ATZ/BEV were similar between both groups. Thus, the efficacies of additional therapies may influence the long-term outcomes. We added the descriptions regarding these matter in Discussion (P15, L25-L34).

4. The analysis in Table 2 may appear somewhat verbose. It would be clear to narrow it down to whether the patients have a history of receiving lenvatinib and conduct the analysis accordingly.

We apologize for our mistake in the title of Table 2b. Table 2a demonstrate factors associated with the therapeutic efficacy assessed based on the best response after 6 and 12 weeks of treatment in a total of 67 patients, while Table 2b demonstrate those in 65 patients consisting of 42 MTA-naïve patients and 23 patients after lenvatinib discontinuation due to PD or adverse events. We revised the title of Table 2b.

5. The percentage of LEN-PD in Figure 2d appears to be incorrect. Also, in Table 2, there are 23 patients who started ATZ/BEV with lenvatinib PD, which is not consistent with the text or FIGURE 2, so please check. Otherwise, the Result section is cumbersome, as noted above. Please reorganize and improve it for clarity.

We apologize for our mistake in the title of Table 2b, which may produce redundancy as pointed out by the reviewer. As mentioned in the above part, the title of Table 2b was revised.

6. There is a mess of "ATZ plus BEV" and "atezolizumab plus bevacizumab" in the text. Please check carefully.

According to suggestion by the reviewer, “ATZ plus BEV” was revised to “atezolizumab plus bevacizumab” throughout the text.

To Reviewer-3

1. The number of patients studied was 82 in total, 33 of whom had been previously treated with MTA and 49 of whom had not been previously treated with MTA. Although the number of cases is large from a single institution, the number of cases is small, and the power of analysis is low to conclude that there is no difference by comparison. In addition, this is a retrospective study, so it is difficult to obtain a definitive conclusion from this study design.

We agree with the comment by the reviewer. Thus, we described the limitation of our study in Discussion (P16, L9-L16) as follows. “First, the study was conducted retrospectively at a single institute, in a small patient cohort. Second, the medium follow-up period in the present study was only 272 days. Although the early therapeutic efficacy and intermediate-term outcome of patients receiving atezolizumab plus bevacizumab were evaluated, the outcomes of the patients over the long term still need to be evaluated. Moreover, most of the patients received varied therapies for HCC following discontinuation of atezolizumab plus bevacizumab, including MTA and TACE/TAE treatments. Therefore, the effects of the heterogeneity of additional therapies after atezolizumab plus bevacizumab therapy on the intermediate-term outcomes of the patients need to be evaluated in a large cohort in the future”.

2. In the comparison of patient backgrounds between the MTA-experienced and MTA-naïve groups, there is a significant difference in the history of TACE or TAI. This could be a major difference in patient background, and the impact of this difference on the results of the analysis cannot be ignored.

According to suggestion by the reviewer, we added “the history of TACE or TAI” as a factor in Tables 2a and 2b and Table 4. Consequently, multivariate analysis revealed that history of MTA therapy with lenvatinib as the factor tended to be associated with an unfavorable efficacy. We revised the descriptions regarding this matter in Results (P7, L24-L26).

3. Although there is a difference in early treatment effect between the MTA-treated and untreated groups, it is necessary to explain why there is no difference in OS in those groups. The multivariate analysis shows that early treatment effect is associated with OS, but this contradicts the conclusion that there is no difference in OS between the two groups.

As pointed out by the reviewer, the OS rates did not differ between MTA-naïve and MTA-experienced patients, while previous history of MTA therapy tended to be associated with unfavorable early therapeutic efficacy and PFS. As shown in Table 1, a percentage of patients with BCLC stage C was higher in the MTA- naïve patients than in the MTA-experienced patients. In contrast, a percentage of patients receiving additional therapies after discontinuation of ATZ/BEV were similar between both groups. Thus, the efficacies of additional therapies may influence the long-term outcomes. We added the descriptions regarding these matter in Discussion (P15, L25-L34).

4. The achievement of CR, PR, or SD is associated with the neutrophil/lymphocyte count ratio in the peripheral blood. Should the neutrophil/lymphocyte count ratio in hepatocellular carcinoma tissue be the same as the ratio in peripheral blood? Also, it is unclear why immune checkpoint inhibitors are more effective when neutrophils outnumber lymphocytes.

To clarify the significance of the N/L ratio, we added Table 3 showing the relation of the N/L ratio and the therapeutic efficacy of ATZ/BEV both in the MTA-naïve patients and MTA-experienced patients. In the present study, however, the ORR did not differ in patients with neutrophil lymphocyte ratio of less than 2.7 as compared to patients with the ratio of 2.7 or more, as shown in Table 3 as well as Table 2, despite that multivariate analysis identified N/L ratio of less than 2.7 as a factor associated with a favorable outcome. These matters were added to the descriptions in Results (P7, L22-L28) and Discission (P15, L21-L23). Also, immunological background responsible for such results should be investigated in future. We described this future problem in Discussion (P15, L23-L24).

5. Please provide the results of the analysis of the relationship between the achievement of CR, PR, or SD and the neutrophil/lymphocyte ratio in the peripheral blood, dividing the patients into two groups: those who have been treated with MTA and those who have not been treated with MTA.

According to suggestion by the reviewer, we added Table 3. Please refer the above answer.

To Editor

1. The reviewer raises significant concerns about the novelty of your study. It is imperative that you clearly differentiate your work from existing literature and articulate the unique contributions of your study, particularly in the context of patients treated with Atezolizumab and Bevacizumab who have experienced MTA.

ATZ/BEV has been used for a lot of patients with previous therapies with MTAs in clinical practice. However, the efficacies and middle-term outcomes, such as PFS and OS, after initiation of atezolizumab plus bevacizumab in the MTA-experienced patients has not yet been elucidated. We evaluated the early therapeutic efficacy, PFS and OS in the MTA-experienced patients, especially lenvatinib-experienced patients. We added the descriptions regarding this matter in Discussion (P14, L26-L29 and P15, L25-L27).

2. The limited number of patients in your study has been noted as a significant limitation. I advise you to acknowledge this explicitly and discuss how this impacts the generalizability of your findings. If possible, provide a justification for the sample size and explain how your study still provides valuable insights despite this limitation.

As described in Discussion (P16, L9-L16), limitations exist in the present study. However, according to suggestion by the editor, we added the description of the early therapeutic efficacy, PFS and OS in the MTA-experienced patients, especially lenvatinib-experienced patients in Discussion (P14, L26-L29 and P15, L25-L27).

3. Please make thorough discussions regarding the clinical impact of ATZ/BEV as second-line therapy, given its current recommendation as a first-line treatment.

Currently, ATZ/BEV is recommended as a first-line therapy for patients who are candidates for systemic chemotherapy. However, ATZ/BEV has been used for a lot of patients with previous treatment with MTAs in clinical practice. Thus, the efficacies of ATZ/BEV as a second-line therapy need to be elucidated. We added this description in Discussion (P14, L26-L29).

4. There are concerns about the clarity and consistency of your data presentation, particularly in Table 2 and Figure 2d. These should be revised for accuracy and coherence.

We apologize for our mistake in the title of Table 2b. Table 2a demonstrate factors associated with the therapeutic efficacy assessed based on the best response after 6 and 12 weeks of treatment in a total of 67 patients, while Table 2b demonstrate those in 65 patients consisting of 42 MTA-naïve patients and 23 patients after lenvatinib discontinuation due to PD or adverse events. We revised the title of Table 2b.

5. The significant difference in patient backgrounds (history of TACE or TAI) between groups needs to be addressed. This could potentially impact the results and should be discussed.

According to suggestion by the reviewer, we added “the history of TACE or TAI” as a factor in Tables 2a and 2b and Table 4. Consequently, multivariate analysis revealed that history of MTA therapy with lenvatinib as the factor tended to be associated with an unfavorable efficacy. We revised the descriptions regarding this matter in Results (P7, L24-L26).

6. Additional analysis is requested regarding the neutrophil/lymphocyte ratio in different patient groups (MTA-treated vs. untreated). This should be included to strengthen your findings.

To clarify the significance of the N/L ratio, we added Table 3 showing the relation of the N/L ratio and the therapeutic efficacy of ATZ/BEV both in the MTA-naïve patients and MTA-experienced patients. In the present study, however, the ORR did not differ in patients with neutrophil lymphocyte ratio of less than 2.7 as compared to patients with the ratio of 2.7 or more, as shown in Table 3 as well as Table 2, despite that multivariate analysis identified N/L ratio of less than 2.7 as a factor associated with a favorable outcome. These matters were added to the descriptions in Results (P7, L22-L28) and Discission (P15, L21-L23). Also, immunological background responsible for such results should be investigated in future. We described this future problem in Discussion (P15, L23-L24).

We thank you and reviewers for their kind suggestions to improve the quality of our manuscript. We hope that our revised manuscript is suitable for publication as an original article in PLOS ONE.

Sincerely,

Shinpei, Yamaba, MD

Satoshi Mochida, MD, PhD

Department of Gastroenterology & Hepatology, Faculty of Medicine, Saitama Medical University

38 Morohongo, Moroyama-cho, Iruma-gun, Saitama 350-0495, Japan.

TEL & FAX: +81-49-276-1198, E-Mail: smochida@saitama-med.ac.jp

---

## [Decision Letter · Decision Letter 1]

31 Jan 2024

Usefulness of Atezolizumab Plus Bevacizumab as Second-line Therapy for Patients with Unresectable Hepatocellular Carcinoma

PONE-D-23-25052R1

Dear Dr. MOCHIDA,

We’re pleased to inform you that your manuscript has been judged scientifically suitable for publication and will be formally accepted for publication once it meets all outstanding technical requirements.

Kind regards,

Jincheng Wang

Academic Editor

PLOS ONE

Additional Editor Comments (optional):

Reviewers' comments:

Reviewer's Responses to Questions

**Comments to the Author**

1. If the authors have adequately addressed your comments raised in a previous round of review and you feel that this manuscript is now acceptable for publication, you may indicate that here to bypass the “Comments to the Author” section, enter your conflict of interest statement in the “Confidential to Editor” section, and submit your "Accept" recommendation.

Reviewer #2: All comments have been addressed

Reviewer #3: All comments have been addressed

2. Is the manuscript technically sound, and do the data support the conclusions?

Reviewer #2: Yes

Reviewer #3: Yes

3. Has the statistical analysis been performed appropriately and rigorously? 

Reviewer #2: Yes

Reviewer #3: Yes

4. Have the authors made all data underlying the findings in their manuscript fully available?

Reviewer #2: Yes

Reviewer #3: Yes

5. Is the manuscript presented in an intelligible fashion and written in standard English?

Reviewer #2: Yes

Reviewer #3: Yes

6. Review Comments to the Author

Reviewer #2: (No Response)

Reviewer #3: The authors provided favorable response. Now the manuscript would be described with technically sounds.

I agreed with the authors' comments.

7. PLOS authors have the option to publish the peer review history of their article (what does this mean?). If published, this will include your full peer review and any attached files.

Reviewer #2: No

Reviewer #3: No
